# Implementing Pharmacogenomics Testing: Single Center Experience at Arkansas Children’s Hospital

**DOI:** 10.3390/jpm11050394

**Published:** 2021-05-11

**Authors:** Pritmohinder S. Gill, Feliciano B. Yu, Patricia A. Porter-Gill, Bobby L. Boyanton, Judy C. Allen, Jason E. Farrar, Aravindhan Veerapandiyan, Parthak Prodhan, Kevin J. Bielamowicz, Elizabeth Sellars, Andrew Burrow, Joshua L. Kennedy, Jeffery L. Clothier, David L. Becton, Don Rule, G. Bradley Schaefer

**Affiliations:** 1Department of Pediatrics, University of Arkansas for Medical Sciences, Little Rock, AR 72202, USA; JEFarrar@uams.edu (J.E.F.); KJBielamowicz2@uams.edu (K.J.B.); EASellars@uams.edu (E.S.); TABurrow@uams.edu (A.B.); KennedyJoshuaL@uams.edu (J.L.K.); BectonDavidL@uams.edu (D.L.B.); SchaeferGB@uams.edu (G.B.S.); 2Arkansas Children’s Research Institute, Little Rock, AR 72202, USA; PortergillPA@archildrens.org (P.A.P.-G.); AVeerapandiyan@uams.edu (A.V.); ProdhanParthak@uams.edu (P.P.); 3Pediatrics and Biomedical Informatics, University of Arkansas for Medical Sciences, Little Rock, AR 72202, USA; Pele.Yu@archildrens.org; 4Departments of Pathology and Laboratory Medicine, University of Arkansas for Medical Sciences and Arkansas Children’s Hospital, Little Rock, AR 72202, USA; bboyanton@uams.edu; 5Informatics, Arkansas Children’s Hospital, 1 Children’s Way #512-10, Little Rock, AR 72202, USA; AllenJC@archildrens.org; 6Pediatric Hematology/Oncology, Arkansas Children’s Hospital, 1 Children’s Way #512-10, Little Rock, AR 72202, USA; 7Pediatric Neurology, Arkansas Children’s Hospital, 1 Children’s Way, Little Rock, AR 72202, USA; 8Pediatric Cardiology, Arkansas Children’s Hospital, Little Rock, AR 72202, USA; 9Section of Genetics and Metabolism, Arkansas Children’s Hospital, Little Rock, AR 72202, USA; 10Psychiatry, College of Medicine, University of Arkansas for Medical Sciences, Little Rock, AR 72205, USA; JLClothier@uams.edu; 11Translational Software, Inc., TSI, Bellevue, WA 98005, USA; Don.rule@translationalsoftware.com; 12Arkansas Children’s Hospital NW, Springdale, AR 72762, USA

**Keywords:** pharmacogenomics (PGx), pediatrics, best practice alerts (BPAs), electronic health records (EHR), genomic indicators, clinical decision support (CDS), phenotype, genotype

## Abstract

Pharmacogenomics (PGx) is a growing field within precision medicine. Testing can help predict adverse events and sub-therapeutic response risks of certain medications. To date, the US FDA lists over 280 drugs which provide biomarker-based dosing guidance for adults and children. At Arkansas Children’s Hospital (ACH), a clinical PGx laboratory-based test was developed and implemented to provide guidance on 66 pediatric medications for genotype-guided dosing. This PGx test consists of 174 single nucleotide polymorphisms (SNPs) targeting 23 clinically actionable PGx genes or gene variants. Individual genotypes are processed to provide per-gene discrete results in star-allele and phenotype format. These results are then integrated into EPIC- EHR. Genomic indicators built into EPIC-EHR provide the source for clinical decision support (CDS) for clinicians, providing genotype-guided dosing.

## 1. Introduction

Adverse drug reactions (ADRs) are the fourth leading cause of death in the USA [1] and account for 135K deaths per year with an economic burden of over USD 136 billion [2,3]. A systematic review of prospective studies showed that 5.3% of hospital admissions were associated with ADRs [4]. These are alarming statistics that illustrate the potential of widespread pharmacogenomic profiling to help mitigate some of these ADRs. More broadly, medical practice is moving away from the concept of “one size fits all” medications [5], as drugs that help some patients will not work for others, and the same drug may have adverse effects in some patients (Figure 1).

Completion of the Human Genome Project [6], International HapMap Project [7], and the 1000 Genomes Project [8] showed the complex nature of underlying human genetic variations, which can determine and contribute to differential drug responses. Pharmacogenomics (PGx) looks at how heritable genetic differences affect individual response to drugs [1,9]. PGx broadly considers an individual’s genetic makeup, lifestyle, and environmental factors to help design interventions that impact drug response and adverse effects. Getting therapeutic choices correct the first time is critical to a successful outcome of drug therapy. Advances in genetics and in our understanding of the potential of the human genome in the pathogenesis of disease and in the prediction of drug treatment effects have spawned a new approach to drug therapy called “genomic medicine”. Genomic medicine has the potential to advance modern therapeutics (e.g., cancer and transplantation treatments) while presenting opportunities to extend value and safety through customized and individualized genotype-guided drug treatments; this is a fundamental premise of precision medicine. Although precision medicine is transforming clinical care for adult medicine, in pediatric medicine this process lags behind for many drugs with well-established pharmacogenetic associations and guidelines [10,11], because the ontogeny of drug metabolizing enzymes and transporters dictates drug response in children and adults.

The 10th Genomic Medicine meeting on “Research Directions in Pharmacogenomics” [12], outlined that success in PGx implementation has been largely through NIH-funded efforts, but recommended broadening these efforts by updating and annotating genomic data in existing electronic health records (EHRs), along with the development of robust “plug-in” modules to make these advances available to the medical practitioner and patient. A successful implementation of PGx in available EHRs requires providing timely information to clinicians in terms of discrete data results, clinical interpretation of phenotype and genotype data, and clinical decision support (CDS) on the PGx actionable variants at the point of care in Epic™ (Epic Systems Corporation, Wausau, WI, USA).

Arkansas Children’s is a medium-large tertiary medical facility providing primary and comprehensive subspecialty care services to children, adolescents, and young adults throughout Arkansas though several campuses and an integrated clinical network of care. Since 2018, the health system has used EPIC (Wausau, WI, USA) as the primary EHR vendor. As dedicated pediatric providers, we believe it imperative that the promise of precision medicine become comprehensively integrated into the pediatric medical care models. With the quality of the current faculty’s expertise, recent recruitment of several highly skilled medical professionals, and expansion of clinical research capabilities, Arkansas Children’s is ideally positioned to quickly become a leader in the field of pediatric precision medicine. Here we describe our early approach to successful PGx implementation in this environment.

## 2. Materials and Methods

A small committee of individuals with expertise in toxicology and genetics convened to form the Precision Medicine (PM) group. Simultaneously, multiple items had to be addressed, including physician and patient family interest, proper instrumentation, genotyping panel design, evaluation of current medications prescribed at the ACH pharmacy, and comprehensive IT support, from genotype calling through EMR reporting and integration. Champion clinicians were enlisted, as well as EPIC support and an outside company to provide robust data interpretation and templated guidance for CDS. A clinical pharmacogenomics (PGx) program was established at Arkansas Children’s Hospital to tailor the therapeutic care delivered to children using genomics. The following project details were reviewed by the Institutional Review Board (IRB) at the University of Arkansas for Medical Sciences (UAMS), Little Rock, Arkansas, which considered the project to be a development and implementation of an internal genetics panel at ACH for the purpose of improving local patient care that, as such, did not meet the regulatory definition of research or require IRB oversight (PI: Schaefer; IRB #-262792).

### 2.1. Adoption of PGx-Patient and Physician Interest Survey

A preliminary electronic survey in August–September 2018 at ACH was given to primary care physicians as well as patients’ families to determine if there was an unmet need to provide PGx testing to improve drug prescribing practices.

### 2.2. Pharmacy Records Data Extraction

For the year 2018 all pharmacy records were evaluated to identify prescribing practices associated with drugs with extant pediatric pharmacogenomic guidance (Table 1) per clinically based guidelines from the U.S. FDA, Clinical Pharmacogenomics Implementation Consortium (CPIC), and the Dutch Pharmacogenetic Working Group (DPWG). Table 1 shows the relevant associated gene/s for the prescribed drugs.

### 2.3. Selection and Generation of PGx Panel

The pediatric PGx test panel was designed to assess genes and gene variants with drug response that were targeted by the above prescribed drugs (Table 1) at ACH and were designated evidence level 1 with established evidence-based clinical guidelines.

#### 2.3.1. Real-Time PCR Instrument and OpenArray^®^ Panel Analytical Validation

The validation of QuantStudio™12K Flex Real-time PCR instrument (Thermo Fisher Scientific, Waltham, MA, USA) was performed in March 2020 by Thermo Fisher Scientific field application specialists. Review of Table 1 gave 23 gene/gene variants targeting 174 clinically actionable SNPs for pediatric genotype-guided drug dosing (Table 2). A Custom OpenArray^®^ (www.ThermoFisher.com) was designed for the 174 SNPs PGx panel and validated by Thermo Fisher Scientific in September 2020.

#### 2.3.2. PGx OpenArray^®^ and CNV Assay Validation

The samples were loaded onto OpenArray^®^ plate using the QuantStudio™12K Flex OpenArray AccuFill System. Detection and genotyping are performed on the QuantStudio™12K Flex system, which includes the QuantStudio Software v1.1.2 and TaqMan^®^ Genotyper Software v1.3. For genotyping TaqMan^®^ assays are used which consist of pre-optimized PCR primer pairs and two probes (FAM dye label and VIC dye label) for allelic discrimination (www.ThermoFisher.com). The complete test consists of 2 components; genotyping and copy number quantification. All genotyping and CNV assays identify alleles in human samples obtained from buccal swabs and blood samples, which are used to determine drug metabolism and specific disease condition risk factors. The copy number component consists of 2 TaqMan copy number assays for the *CYP2D6* gene (exon 9 and intron 2).

### 2.4. Build of Genomic Indicators in EPIC

One of the challenging aspects of implementing pharmacogenetics across systems is that there is no standardized nomenclature to draw upon. While simple star-allele labels have gained acceptance for diplotypes (combinations of haplotypes), there are a wide variety of ways to represent the complex structural variations inherent in the *CYP2D6* gene. Although normal, intermediate, and rapid metabolizer monikers have become common for phenotypes, the terms “poor” or “intermediate” metabolizer do not capture subtle variations in *CYP2C9* metabolism that are indicated in current CPIC guidelines. To provide a more granular view of metabolism, a growing number of recommendations are provided on the basis of the predicted activity score of the resulting enzyme. Even the labeling of individual single nucleotide polymorphisms (SNPs) will be different when test results are analyzed with different genome builds. To alleviate these issues, Translational Software, Inc. (TSI, Bellevue, WA 98005, USA) collaborated with EPIC (Wausau, WI, USA) to establish a numbering system that represents both the type of result that is reported as well as the specific result for genotypes, diplotypes, phenotypes, and activity scores for each allele/SNP on the ACH PGx Panel. These numbers become the basis for rules that determine which specific recommendations to provide to the clinician as best practice alerts (BPAs). In the example (Table 3) below, a clinician would be warned that PGx test results indicate an increased risk of adverse effects or therapeutic failure for Amitryptyline. The recommendation would be to consider an alternative medication or use therapeutic drug monitoring to guide Amitryptyline dose adjustments. The reason for that guidance is based upon the patient’s *CYP2C19* and *CYP2D6* metabolizer test results, which show a poor metabolizer phenotype for *CYP2C19* with diplotypes (**2/*2*, **2/*3*, or **3/*3*); and a rapid metabolizer phenotype for *CYP2D6* with diplotypes (**1/**1)xN, (**1/*2*)xN, or (**2/**2)xN [13]. Note that Table 3 shows two distinct identifiers that are labeled and may result in different recommendations for other drugs.

## 3. Results

### 3.1. Pharmacogenomics Program at ACH

#### 3.1.1. Patient and Physician Interest Survey

The preliminary survey of physicians and patients’ families (*unpublished*) showed that 88% of patients’ parents (*n* = 49) were interested in having their child’s DNA used to guide medical diagnosis and drug therapy. By contrast, 81% of ACH physicians (*n* = 206) indicated an interest in some kind of warning of drug-gene interaction within a patient’s EHR. It was determined that there was a clear interest by the stakeholders (clinicians and patient families) and the PM Group in the development and implementation of a PGx program at ACH that would be of high priority and allow us to provide personalized drug therapy to our patients at ACH.

#### 3.1.2. ACH Pharmacy Records Review

A review of pharmacy records for the year 2018 showed that 42,877 prescriptions (in-patient and out-patient) were filled by the hospital (Table 1). The pediatric PGx test panel covers the most-prescribed medications at ACH and was coupled with pediatric clinical guidelines to improve patient care. For example, prescriptions of ondansetron were filled 21,147 times followed by oxycodone at 12,978 times. These commonly prescribed medications covered 11 medical specialties including anesthesia, cancer, cardiology, gastrointestinal, genetics, hematology, infectious diseases, neurology, pain, psychiatry and addiction, and transplantation (Table 4). The above-listed personalized medications were explored for their pharmacogenetic data and clinical annotations in the U.S. FDA, Clinical Pharmacogenomics Implementation Consortium (CPIC), and the Dutch Pharmacogenetic Working Group (DPWG) and PharmGKB [14,15,16,17]. Our search gave us 23 actionable PGx variants and their associated pathogenic variant single nucleotide polymorphisms (SNPs), for a total of 174 SNPs (Table 2), that have drug response phenotypes.

#### 3.1.3. PGx Assay Performance

Laboratory developed tests (LDTs) [18,19] including PGx testing, are developed and implemented to fulfill the unmet medical needs of an institution and the patient population it serves, when such laboratory-based testing is not commercially available. Although LDTs do not necessarily require approval from the FDA, laboratories must, at minimum, adhere to regulations set forth by the Clinical Laboratory Improvement Amendments of 1988 (CLIA’88) [20]. Thus, it is essential to thoroughly validate the performance of a laboratory method and establish standard operating procedures prior to implementing the test for clinical purposes.

The ACH PGx test was designed for a high-throughput laboratory that can process hundreds of samples across a large number of targets. For analytical validation, we used 24 Coriell samples and 10 different plasmid pools. Details could be sent to the reader upon request for critical components of the validation of this qualitative genotyping test, establishing the DNA sample concentration dynamic range, reproducibility, accuracy of genotyping, and copy number variation (CNV). Briefly, the dynamic range of tested DNA sample concentrations was between 3.13 ng/µL and 25 ng/µL. For CNV, good results were obtained for a final concentration between 10 ng/reaction (5 ng/µL) and 5 ng/reaction (2.5 ng/µL). Accuracy of genotyping and CNV results were 99.97% and 95.55%, respectively. Overall genotyping and CNV reproducibility were 99.52% and 99.26%, respectively. Finally, verification of the PGx test’s ability to correctly detect polymorphisms was assessed using a mixed pool of samples (plasmid pool controls from Coriell (Camden, NJ, USA) and volunteer human samples (blood and buccal DNA)); all test results were specific to their intended SNP target. In addition, we performed bi-directional sequencing of volunteer DNA samples and thereby confirmed genotype data generated from the OpenArray PGx test (Table 2).

#### 3.1.4. Integrating PGx into EHR with CDS

A significant impediment to the adoption of pharmacogenetics has been an inability to incorporate real-time physician education, CDS, and active ordering of PGx-directed genetic testing into the physician’s normal workflow in EPIC. Under the guidance of ACH, EPIC collaborated with Translational Software, Inc. (TSI) to import TSI’s rule set and curated recommendations into best practice advisories (BPAs). With this content enabled, physicians are prompted to consider PGx testing as per pre-alerts (Figure 2A,B) that are triggered by prescription orders of PGx relevant drugs.

When physicians order the PGx test in EPIC, there is an option to select drug-gene pair from the listed conditions (for example, cancer, cardiovascular, etc.) (Figure 2A); the system then prompts the physician to obtain genetic consent from the patient/family member and also to obtain PGx test pre-authorization (Figure 2B). If the pre-authorization is not available, then the patient blood sample can only be processed for DNA extraction. When the pre-authorization is received, the DNA sample is automatically placed for PGx panel runs in the molecular pathology laboratory at ACH. SNP data is reviewed by a molecular pathologist, then sent to TSI via a secured cloud platform using HL7/SFTP interphase. When SNP data results are completed, they are processed by TSI’s cloud-based platform to provide a summary report as well as discrete test results that are submitted to EPIC’s Advanced Genomic Module (Figure 3). The summary report provides the ordering physician an overview of the implications of the genetic test for the patient, and the test results in the Advanced Genomic Module provide data in a computable format that enables perpetual use of the test results within the EHR system. Once the test results are received, subsequent medication orders are evaluated on the basis of the rule set that is integrated into EPIC using component codes for gene, genotype, and phenotype of interest provided by the ACH-IT PGx group. When there are potential PGx implications for medication orders prepared by the physician, best practice advisories (BPAs) that provide evidence-based recommendations, including the clinical guidelines of the U.S. FDA, CPIC, and DPWG, are triggered.

The specific drug-gene pairs that are the focus of the alerts are high-risk PGx actionable genotypes (Figure 4A–C) and has been carefully chosen to avoid “alert fatigue” on the part of clinicians. The early implementation strategy at ACH is to focus on a handful of subspecialties that are either highly receptive to (e.g., genetics) or familiar with using PGx on some limited basis (e.g., oncology, neurology, and cardiology) to build out the process and scrutinize the BPAs. The goal is to grow to high-volume primary care clinics. BPAs may be passive or interruptive and may be configured to suggest alternative medication orders and enable the clinician to indicate why they are maintaining the current order.

#### 3.1.5. Examples of BPAs in EPIC

We initially identified a small group of champion physicians for PGx implementation, including those with either a strong intrinsic interest in PGx (e.g., genetics) or who were thought to be most easily capable of incorporating PGx into their clinical practice work flows. Champion physicians for cardiovascular, neurology, hematology, and oncology were identified, and internal group meetings with the respective physician teams arranged. These clinicians reviewed and in some cases adjusted the BPA language to best fit local standards and practice in the EPIC-based EHR for genotype-guided therapy. Below are examples of interruptive BPAs for clopidogrel, thiopurine, and phenytoin (Figure 4A–C) (EPIC © 2021 Epic Systems Corporation). As mentioned earlier, we established a numbering system (Table 3) that provides CDSs for specific discrete results for 174 SNPs targeting 66 pediatric drugs on the ACH PGx Panel. The alert provides a tab where clinician can review a patient’s full list of genomic indicators. Below are some examples of BPAs that will be invoked in EPIC for high-risk PGx actionable genotypes, when clinicians receive genotype-guided recommendations for a particular pediatric medication:

Clopidogrel-*CYP2C19*

A patient with a poor metabolizer phenotype for *CYP2C19* could have diplotypes as **2/*2*, **2/*3* or **3/*3* [21]. BPA as shown in Figure 4A will be triggered, letting the physician know that the phenotype and genotype results for *CYP2C19* show significantly reduced response to clopidogrel and consider antiplatelet agents (prasugrel, ticagrelor) as a therapy alternative.

Thiopurine-*TPMT/NUDT15*

If a patient has a poor metabolizer status for *TPMT* with diplotypes (**3A/*3A*, **2/*3A*, **3A/*3C*, **3C/*4*, **2/*3C*, **3A/*4*) and normal metabolizer for *NUDT15* (*1/*1) [22], the BPA for mercaptopurine shown in Figure 4B will trigger. This alerts the physician that the patient has increased risk for myelotoxicity and that the genotype results predict life-threatening risk of leukopenia, neutropenia, and myelosuppression with standard doses of mercaptopurine.

Phenytoin-*CYP2C9*

If the discrete result shows a patient has intermediate metabolizer status for *CYP2C9* phenotype, with associated diplotypes **1/*2*, **1/*3* or **2/*2* [23], then the alert in Figure 4C will trigger, showing that patients with *CYP2C9* intermediate phenotype and taking phenytoin are at increased risk of mild to moderate neurological toxicity.

In addition, to complement the EHR-intrinsic features of the return of PGx results to physicians through lab review modules and BPAs, we also formulated a comprehensive PGx report as a PDF document (Figure 3) showing details on current patient medications, risk management, potentially impacted medications, dosing guidance on drug-gene interactions, drug-drug interactions and test details. Appendix A displays a mock report for an patient on Clopidogrel and Codeine. This PGx report shows the clinician the *CYP2C19* and *CYP2D6* metabolizer status and dosing recommendations for adults as well as pediatric patients (Appendix A). Appendix A on the other hand, provides the clinician with a broader examination and understanding of the 23 PGx genes along with their genotype and phenotype status.

At ACH, we have implemented a pediatric personalized medicine program in PGx with ACH-IT focused on data management that is tied directly with electronic health records (EHRs) to alert physicians about drug-gene information that could aid in drug treatment decisions.

## 4. Discussion

Pediatric adverse drug reactions (ADRs) are a significant health concern [24,25], and clinical implementation of pharmacogenomics (PGx) may see the earliest and broadest use in clinical practice for improving patient care [26]. Pediatric drug studies were instrumental in adding dosing and risk information to the labelling of 80 pediatric drugs [27]. Ontogeny controls the pharmacokinetics and pharmacodynamics of drug response [28,29] and is the key to understanding the variability of drug efficacy/toxicity in neonates, adolescents, and adults. The cytochrome P-450 family of enzymes undergoes substantial ontogenic changes. For example, enzyme activity for *CYP2C19*, *CYP2C9*, *CYP3A4*, and *CYP2D6* until birth is low, but reaches adult levels in the first few weeks or months after birth [30]. Similarly, for *TPMT*, genetic polymorphism is a significant factor responsible for serious ADRs (myelosuppression) in patients treated with thiopurines, and *TPMT* activity is higher in infants and children than in adults when normalized for genotype [31]. Health care systems are slowly beginning to implement PGx tests that are at the forefront of moving precision medicine/genomic medicine to a new level, but there is still an urgent unmet need in pediatrics to refine and develop precision actionable PGx guidance through cutting-edge clinical research.

### 4.1. Clinical Utility of Pharmacogenomics

The majority of pharmacogenomic marker associations are based on the progress made in understanding the clinically actionable PGx variants in adult patient populations [15,16,32,33]. Codeine is a prodrug dependent on *CYP2D6* activation, and a majority of PGx guidance for it comes from adult studies [10,34]. Black box warnings on codeine were applied in 2013 and 2017 for children younger than 12 years of age [34], as PGx evidence showed codeine can cause respiratory depression and death. For drug-gene interaction, several drugs now available target PGx variants for which pediatric clinical guidelines are recommended [34]. The PGx test at ACH consists of 174 variant alleles in 23 pharmacogenes (Table 2) and can give guidance on 66 of the most commonly prescribed pediatric drugs. The list of personalized medications in recent years that have pharmacogenomic guidance has increased to 286 [35]. A clinician has the ability to look at a patient’s genetic profile to determine if the treatment options will benefit the patient. Consortia such as CPIC and DPWG provide pharmacogenomic-based evidence guidelines for drug-gene pairs, along with data on frequency of polymorphisms in ethnic groups and their allele functionality status.

### 4.2. PGx Programs at Other Pediatric Medical Centers

Precision medicine (PM) is more common in adult medicine. Because children are our future, it is imperative that PM be integrated into ACH’s method of clinical care. Assessing the pediatric patient as an individual will provide the best and most effective medical treatments.

At the beginning of the evolution of PGx at ACH, our group evaluated PM groups at other US and Canadian pediatric hospitals, along with their pharmacogenomic testing capabilities. We found that very few offer in-house PGx testing for children. Some pediatric hospitals offer PGx testing, but it is out-sourced to institutions such as OneOme (Minneapolis, MN, USA), ARUP (Salt Lake City, UT, USA), Gene by Gene (Houston, TX, USA), and RPRD (Milwaukee, WI, USA). In situations such as this, the interpretation of the PGx test is left to the out-sourced testing company. In some cases, in-house staff are necessary to give the interpretation data a second look, and to manually import the PGx test results into their hospital’s electronic health record (EHR) system. These out-sourced PGx tests are not always pediatric specific, and adult drug:gene assays are included in the results. In addition, interpretations of the PGx results are not populated in the patients’ EHRs as discrete information. The reader is advised to examine the CPIC implementation website for the institutes and commercial entities that utilize CPIC guidelines for PGx testing (https://cpicpgx.org/implementation/, accessed on 1 March 2021).

It is financially beneficial that ACH can offer PGx testing for its pediatric patients, and that all assays on the PGx panel have pediatric guidelines. Our PM group has one doctor of pharmacy that can interact with providers at any time to offer education and clinical information support. Our testing facility is housed within the Department of Pathology and Laboratory Medicine, which conforms to all regulations mandated by CLIA ‘88, the College of American Pathologists, and the American Association of Blood Banks. The accuracy of all analytical data is verified by a board-certified molecular pathologist, then sent to a third party for final genotype-phenotype interpretation. St. Jude’s and Cincinnati Children’s offer excellent PGx testing services comparable to the testing at ACH. For a PGx test to be successful, it requires complete patient drug coverage, easy ordering by providers, a quick turnaround time for interpreted results, and all easily available in the patient EHR.

### 4.3. EHR-Based Clinical Decision Support Systems (CDSS)

Pediatric hospitals have begun integrating PGx CDSS in their EHRs. The level of integration includes real-time at the point of care PGx treatment guidelines and dosing recommendations, ranging from interruptive alerts suggesting treatment recommendations to providing guidance to the presence of a specific PGx variant for a specific medication [11,36,37]. Some institutions leverage the use of machine learning and natural language processing to extract triggers for CDSS [38], and others have incorporated the patient consent process into the CDS workflow [36]. More importantly, institutions have established CDSS governance structures to help guide the implementation of PGx CDSS [11,36,37].

### 4.4. Challenges and Barriers to PGx Implementation

An extensive review of ongoing clinical pharmacogenomics implementation programs at various hospitals and institutions highlighted several adoption and implementation barriers for us to overcome, including scientific, information technology (IT), lack of education of clinical staff and patients, test reimbursement, PGx clinical decision support (CDS), lack of clinician adoption, and data storage for clinical research [12,39,40,41,42].

#### 4.4.1. Education of Future Clinicians on PGx

A recent survey [43] of medical schools about perceptions on adding pharmacogenomics instruction to the medical curricula found that physicians and health care workers do not possess appropriate knowledge of PGx. A more recent work also illustrated the point that pharmacists and clinicians can gain understanding of PGx through education [44]. There is an urgent need to incorporate the PGx curriculum in medical school education, so the next generation of physicians can incorporate personalized therapies for their patients’ wellbeing into their routine clinical practice.

In addition, much has been evaluated about the role of pharmacists in PGx. In a recent study, 35% (339 out of 978) of last-year pharmacy students across eight east coast college’s felt that pharmacogenomics is a useful tool for pharmacists, yet only 40% of these same students considered it to be an important part of their training [45]. There was varying exposure to pharmacogenomics training, although many understood the clinical importance of PGx. At ACH, the plan is to focus on subspecialties that are knowledgeable of PGx and then expand to other subspecialties by building BPAs and educating clinical staff.

#### 4.4.2. Consent/Parent Awareness

An important component of PGx testing at ACH is the informed consent process. With this in mind, it is crucial that the family be aware that the testing is offered and ordered, as appropriate. Parents/Guardians and/or patients must be given complete information about the risks and benefits of this testing and be given the opportunity to ask questions before providing informed consent. As with other genetic testing consent procedures, this should occur in an area that allows the patient and/or their parents and guardians to focus on the information presented without overt distractions. Empowering parents/guardians and patients as crucial members of the care team provides the opportunity for them to advocate for their health as future need arises.

#### 4.4.3. Cost of PGx Test

The configuration of the OpenArray^®^ (www.Thermofisher.com) TaqMan SNP-based genotyping PGx panel can accommodate a maximum of 16 samples per run; of these, two are dedicated to quality control. Therefore, a maximum of 14 patient samples can be performed on a single open-array chip. Fixed-cost test components include all reagents needed to extract and purify genomic DNA and perform analytical testing (USD 170 per patient sample), and the interpretative report generated by Translational Software Inc. (USD 35 per patient sample). Variable cost test components include technical labor (9 h at USD 30 per hour) and the OpenArray^®^ chip (USD 720). More specifically, the variable cost of technical labor ranges from USD 270 (1 patient sample per run) to USD 19 (14 patient samples per run). Likewise, the variable cost of the OpenArray^®^ chip ranges from USD 720 (1 patient sample per run) to USD 51 (14 samples per run). Using a blended fixed and variable cost matrix calculation, the total cost to perform the open array pharmacogenomics panel ranges from USD 1195 (1 patient plus 2 controls) to USD 276 (14 patients plus 2 controls).

#### 4.4.4. Test Reimbursement

For billing purposes, each gene on the SNP-based genotyping pharmacogenomics test is paired with the corresponding AMA-approved CPT code as follows: *CYP2C19* (81225), *CYP2D6* (81226), *CYP2C9* (81227), *CYP3A4* (81230), *CYP3A5* (81231), *F2* (81240), *F5* (81241), *G6PD* (81247), *NUDT15* (81306), *SLCO1B1* (81328), *TPMT* (81335), *UGT1A1* (81350), and *VKORC1* (81355). CPT code 81479 is used for the following genes that currently lack AMA-approved CPT codes: *ACYP2*, *CEP72*, *CYP2C*, *CYP4F2*, *RARG*, *SLC6A4*, *SLC28A3*, *RYR1*, and *CACNA1S*. From 21 September 2020 to 12 March 2021, a total of 29 tests had been ordered. At this time, we do not have sufficient data to understand reimbursement based upon patient insurance (Medicaid, private insurance, etc.).

### 4.5. Integration of PGx Test Results into EPIC

Integration of PGx data into the EHR must ensure that the information related to the drug-gene pair maintains analytical validity, as well as the clinical utility of the test [46]. Currently, clinically relevant genetic results are often considered very similar to laboratory results, where the genetic report and raw genetic data are stored outside of the EHR [47]. Standards in implementing, storing, and transmitting genetic information across EHR systems have been poorly adopted. When it contains properly integrated PGx data, an EHR should be able to support timely access to genomic information at the point of care, trigger clinical decision support mechanisms, and facilitate ordering tests and tracking their results as well as notifying patients and families [48]. A recent survey of multidisciplinary healthcare providers found that 71.3% were slightly or not at all familiar with PGx, which suggests additional education and electronic resources are needed for pediatric PGx examples [49].

We performed a pharmacogenomics knowledge-assessment survey of prescribers at Arkansas Children’s prior to the launch of PGx testing to help tailor our educational program [50]. The survey showed that prescribing clinicians are interested in the opportunity to provide PGx testing to their patients at ACH. Prescribers recognized the need for additional information about PGx and welcomed eLearning and specialty-specific educational sessions as alternative means of education. In addition, clinicians were concerned about cost, turnaround time, and efficacy of the test. Of note, a representative sample of younger clinicians (resident house staff) responded to the survey, perhaps presenting as a marker for their readiness to consider PGx as part of their routine decision-making process.

## 5. Conclusions

Pharmacogenomics (PGx) can help prevent ADRs and improve drug efficacy by enabling the physician to optimize drug dosage and avoid prescribing medications with adverse reactions due to the patient’s genetic makeup. At ACH, PGx testing was successfully implemented with EPIC-based clinical decision support (CDS) for 66 pediatric drugs based upon genotype analysis of 174 single nucleotide polymorphisms (SNPs) targeting 23 actionable PGx gene variants. The clinicians receive discrete results for genotype-guided therapy in EPIC-based EHR. Although laboratory turnaround times are relatively short, it is not unusual for a patient’s medication list to change in the time between when a test is ordered and the report is generated. To make the final PGx report more accurate, an application is in development under EPIC’s App Orchard program that will allow the reporting engine to query the patient’s medication list using EPIC’s implementation of the Fast Health Interoperability Resource (FHIR) and an application programming interface (API). Efforts are underway to educate providers on how to order and incorporate these data into standard practice. We hope to provide cutting-edge technology and knowledge to a pediatric population that is often forgotten, but we know this starts with understanding the data and educating team members.

Pharmacogenomics is fast becoming a mainstay for the delivery of 21st century healthcare. Arkansas Children’s clinicians are open to learning more about the promise of PGx for genotype-guided dosing and are eager to utilize this process to improve the quality of pediatric clinical care.

## Figures and Tables

**Figure 1 jpm-11-00394-f001:**
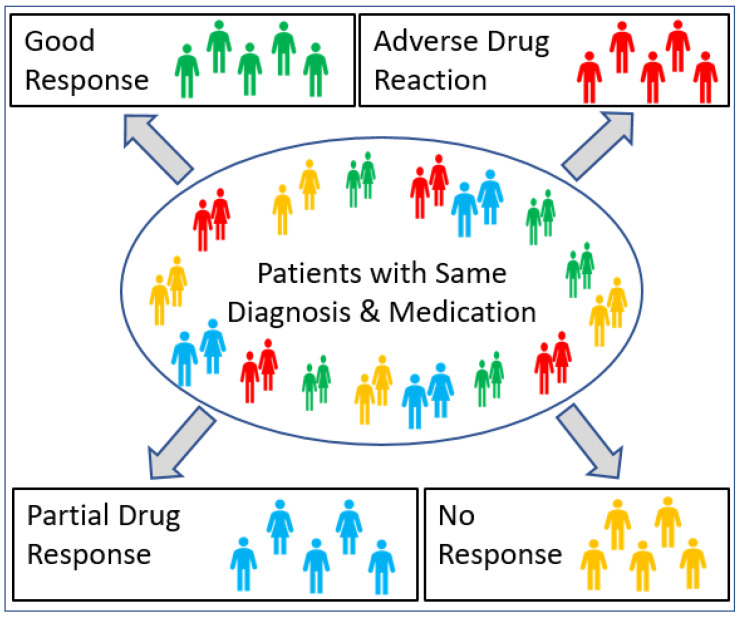
Pharmacogenomics and drug response in individuals with different genotypes.

**Figure 2 jpm-11-00394-f002:**
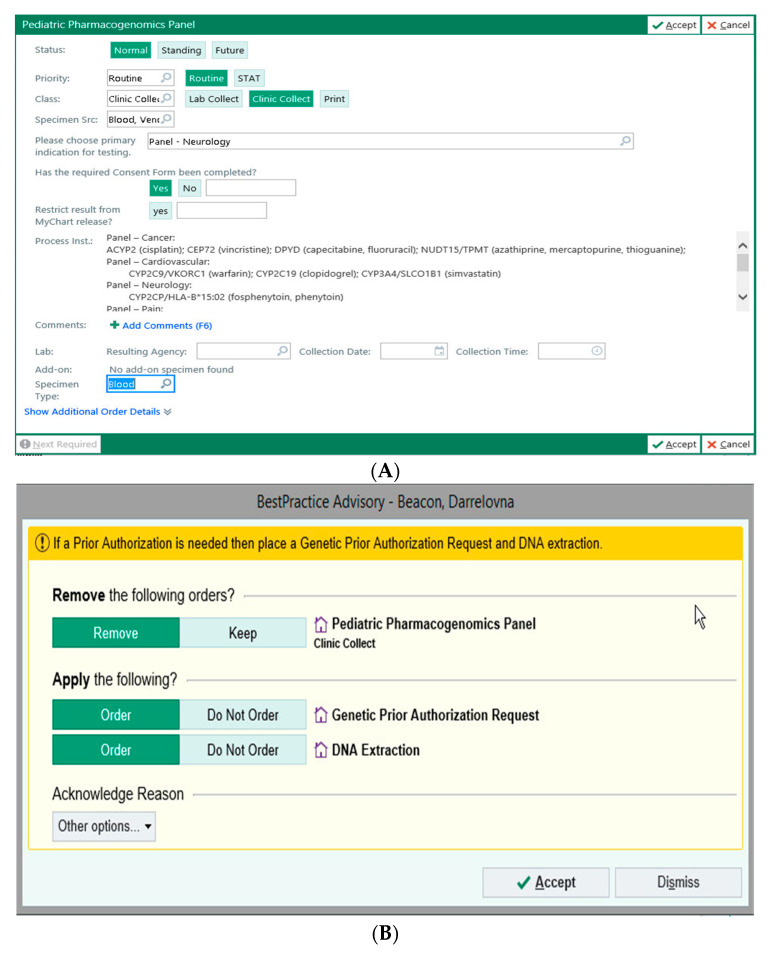
Placing an order for PGx test in ACH EPIC (**A**,**B**).

**Figure 3 jpm-11-00394-f003:**
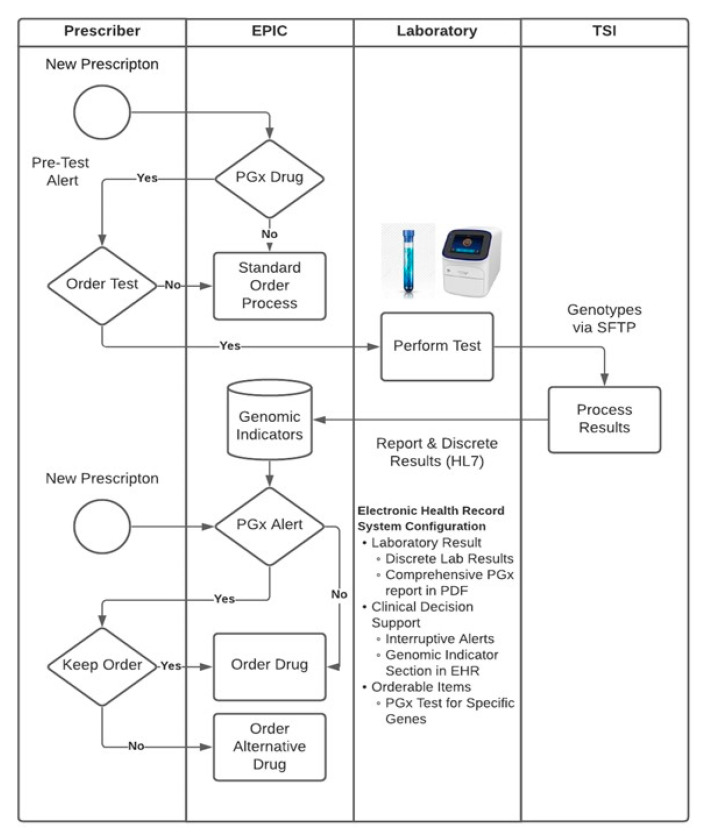
Implementation of pharmacogenomic workflow at ACH. The clinician can order the PGx test in EPIC; genomic indicators in EPIC were built with respective clinical decision supports (CDS). Physicians can receive discrete laboratory results, comprehensive PGx report, and BPAs.

**Figure 4 jpm-11-00394-f004:**
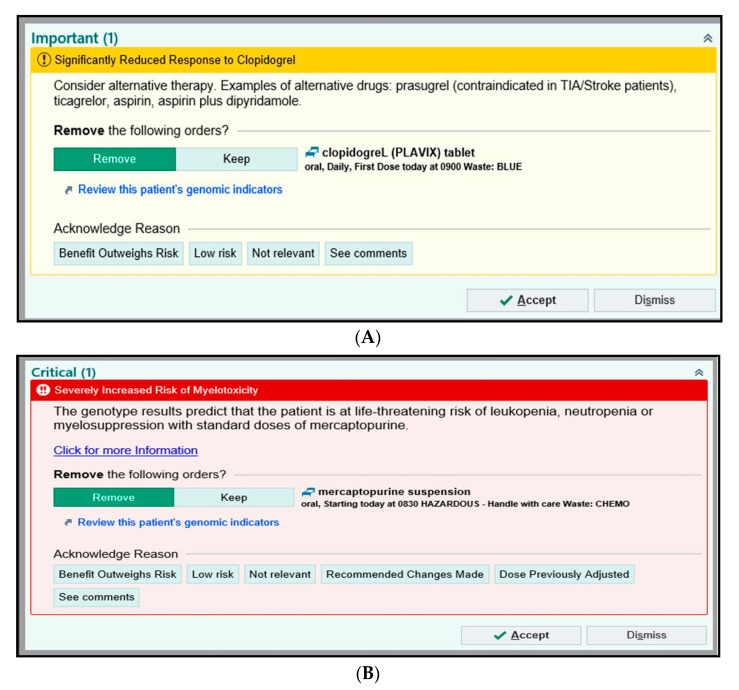
Best practice alerts (BPAs) for high risk PGx actionable genotypes with clinical decision support (CDS) for clinicians in EPIC (EPIC © 2021 Epic Systems Corporation): (**A**) clopidogrel, (**B**) mercaptopurine and (**C**) phenytoin.

**Table 1 jpm-11-00394-t001:** Prescriptions (in-patient and out-patient) filled at ACH Pharmacy in 2018.

Drug Name	Gene/s	Total	Drug Name	Gene/s	Total
Amikacin	*MT-RNR1*	6	Hydrocodone	*CYP2D6*	303
Amitriptyline	*CYP2C19*, *CYP2D6*	1382	Imipramine	*CYP2C19*, *CYP2D6*	190
Aripiprazole	*CYP2D6*	24	Mercaptopurine	*TPMT*, *NUDT15*	809
Atazanavir	*UGT1A1*	3	Neomycin	*MT-RNR1*	33
Atomoxetine	*CYP2D6*	59	Nortriptyline	*CYP2D6*	147
Azathioprine	*TPMT*, *NUDT15*	116	Ondansetron	*CYP2D6*	21,147
Celecoxib	*CYP2C9*	55	Oxybutynin	*NA*	795
Cisplatin	*ACYP2*	59	Oxycodone	*CYP2D6*	12,978
Citalopram	*CYP2C19*	21	Paroxetine	*CYP2D6*	5
Clomipramine	*CYP2C19*, *CYP2D6*	1	Phenytoin	*CYP2C9*	31
Clopidogrel	*CYP2C19*	16	Pimozide	*CYP2D6*	9
Doxepin	*CYP2C19*, *CYP2D6*	11	Salmeterol	*ADRB2*	224
Eltrombopag	*F5*	7	Sertraline	*CYP2C19*	271
Escitalopram	*CYP2C19*	46	Simvastatin	*CYP3A4*, *SLCO1B1*	24
Fluorouracil	*DDYD*	9	Tacrolimus	*CYP3A5*	463
Fluoxetine	*CYP2D6*	197	Thioguanine	*TPMT*, *NUDT15*	36
Fluvoxamine	*CYP2D6*	11	Tobramycin	*MT-RNR1*	61
Formoterol	*ADRB2*	995	Tramadol	*CYP2D6*	90
Fosphenytoin	*CYP2C9*	113	Vincristine	*CEP72*	1528
Gentamicin	*MT-RNR1*	258	Voriconazole	*CYP2C19*	31
			Warfarin	*CYP2C9*, *VKORC1*, *CYP2C*, *DYP4F2*	313

**Table 2 jpm-11-00394-t002:** ACH pharmacogenomics (PGx) panel design summary highlighting gene, number of SNPs and SNP Rs#.

PGx 174 SNP Panel		
Gene	No. of SNPs	SNP rs#
*ACYP2*	1	rs1872328
*CACNA1S*	2	rs772226819, rs1800559
*CEP72*	1	rs924607
*CYP2C*	1	rs12777823
*CYP2C19*	12	rs12769205, rs12248560, rs17884712, rs72552267, rs4986893, rs56337013, rs72558186, rs6413438, rs58973490, rs41291556, rs28399504, rs4244285
*CYP2C9*	13	rs72558193, rs72558189, rs2256871, rs7900194, rs1799853, rs1057910, rs28371686, rs9332239, rs56165452, rs28371685, rs9332131, rs72558187, rs72558190
*CYP2D6*	42	rs730882170, rs28371710, rs1135822, rs267608319, rs28371696, rs267608279, rs16947, rs35742686, rs72549352, rs61736512adjC, rs61736512, rs148769737, rs148769737, rs267608297, rs267608313, rs28371706, rs1065852, rs1135840, rs3892097, rs769258, rs5030862, rs201377835, rs5030867, hCV32407220, rs72549349, rs5030656, rs72549351, rs72549353, rs28371717, rs72549356, rs5030655, rs774671100, rs1080985, rs59421388, rs72549348, rs28371725, rs72549346, rs72549347, rs1135823, rs5030865, rs5030865, rs730882251
*CYP3A4*	5	rs4986910, rs4987161, rs12721629, rs55785340, rs35599367
*CYP3A5*	3	rs776746, rs10264272, rs41303343
*CYP4F2*	1	rs2108622
*DPYD*	22	rs75017182, rs3918289, rs3918289, rs1801159, rs1801158, rs1801268, rs1801267, rs1801266, rs1801265, rs1801160, rs55886062, rs2297595, rs17376848, rs56038477, rs67376798, rs6670886, rs3918290, rs72549309, rs72549306, rs72549310, rs80081766, rs115232898
*F2*	1	rs1799963
*F5*	1	rs6025
*G6PD*	11	rs137852328, rs72554665, rs72554665, rs137852328, rs78478128, rs1050828, rs1050829, rs5030868, rs137852339, rs76723693, rs5030869
*NUDT15*	3	rs766023281, rs116855232, s186364861
*RARG*	1	rs2229774
*RYR1*	41	rs118192163, rs28933396, rs118192176, rs193922770, rs144336148, rs118192162, rs118192116, rs121918592, rs1801086, rs118192161, rs28933397, rs63749869, rs121918594, rs118192170, rs193922802, rs118192167, rs121918595, rs118192168, rs121918593, rs118192124, rs118192122, rs118192175, rs118192172, rs111888148, rs112563513, rs118192178, rs121918596, rs193922747, rs193922748, rs193922753, rs193922764, rs193922768, rs193922772, rs193922803, rs193922807, rs193922816, rs193922818, rs193922832, rs193922843, rs193922876, rs193922878
*SLC28A3*	1	rs7853758
*SLCO1B1*	1	rs4149056
*TPMT*	5	rs1142345, rs56161402, rs1800584, rs1800462, rs1800460
*UGT1A1*	3	rs4148323, rs35350960, rs887829
*UGT1A6*	1	rs17863783
*VKORC1*	1	rs9923231

SNP = single nucleotide polymorphism; CNV assays: *CYP2D6*—Hs00010001_cn (exon 9) and Hs04502391_cn (Intron 6); All alleles that are negative for above sequence variation were defaulted to *1 assignment.

**Table 3 jpm-11-00394-t003:** Example for Amitriptyline illustrates a numbering system designed and built in EPIC and showing the specific result for genotypes, diplotypes, phenotypes, and activity scores for each SNP on the ACH PGx panel.

ID	Title	Indicators	Ordinal Value	Meaning	Logic
882001281	Decreased Amitriptyline Exposure	87005959	1	Amitryptyline Prescribed	1 AND ((2 AND 3) OR (3 AND 4))
		873000170	2	*CYP2C19* Poor Metabolizer	
		873000181	3	*CYP2D6* Rapid Metabolizer	
		873001569	4	*CYP2C19* Poor Metabolizer	

**Table 4 jpm-11-00394-t004:** ACH pharmacogenomics panel and drug-gene targets by pediatric therapeutic area. Bold drugs show only adult guidance.

Condition	Drug	Gene/s
Anesthesia	Desflurane	*RYR1/CACNA1S*
	Enflurane	*RYR1/CACNA1S*
	Halothane	*RYR1/CACNA1S*
	Isoflurane	*RYR1/CACNA1S*
	Sevoflurane	*RYR1/CACNA1S*
	Succinycholine	*RYR1/CACNA1S*
Cancer	Azathioprine	*TPMT/NUDT15*
	Capecitabine	*DPYD*
	Cisplatin	*ACYP2*
	Daunorubicin	*RARG/UGT1A6/SLC28A3*
	Doxorubicin	*RARG/UGT1A6/SLC28A3*
	Fluorouracil	*DPYD*
	Mercaptopurine	*TPMT/NUDT15*
	**Rasburicase**	*G6PD*
	Thioguanine	*TPMT/NUDT15*
	Vincristine	*CEP72*
Cardiovascular	Clopidogrel	*CYP2C19*
	Propranolol	*CYP2D6*
	Simvastatin	*CYP3A4/SLCO1B1*
	Warfarin	*CYP2C9/VKORC1/CYP2C/CYP4F2*
Gastrointestinal (GI)	Dexlanosoprazole	*CYP2C19*
	Esomeprazole	*CYP2C19*
	Lansoprazole	*CYP2C19*
	Omeprazole	*CYP2C19*
	Ondansetron	*CYP2D6*
	Pantoprazole	*CYP2C19*
	Rabeprazole	*CYP2C19*
Gaucher Disease	Eliglustat	*CYP2D6*
Hematology	Eltrombopag	*F5*
Infectious Disease	Atazanavir	*UGT1A1*
	**Chloroquine**	*G6PD*
	Dapsone	*G6PD*
	**Nitrofurantoin**	*G6PD*
	**Primaquine**	*G6PD*
	**Proguanil**	*CYP2c19*
	**Quinine**	*G6PD*
	**Sulfamethoxazole**	*G6PD*
	**Tafenoquine**	*G6PD*
	Voriconazole	*CYP2C19*
Neurology	Fosphenytoin	*CYP2C9*
	Phenytoin	*CYP2C9*
Pain	Celecoxib	*CYP2C9*
	Codeine	*CYP2D6*
	Hydrocodone	*CYP2D6*
	Ibuprofen	*CYP2C9*
	Meloxicam	*CYP2C9*
	Oxycodone	*CYP2D6*
	Tramadol	*CYP2D6*
Psychiatry and Addiction Medicine	Amitriptyline	*CYP2C19/CYP2D6*
	Aripiprazole	*CYP2D6*
	Atomoxetine	*CYP2D6*
	Citalopram	*CYP2C19*
	Clomipramine	*CYP2C19/CYP2D6*
	Desipramine	*CYP2D6*
	Doxepin	*CYP2C19/CYP2D6*
	Escitalopram	*CYP2C19*
	Fluoxetine	*CYP2D6*
	Fluvoxamine	*CYP2D6*
	Iloperidone	*CYP2D6*
	Imipramine	*CYP2C19/CYP2D6*
	Nortriptyline	*CYP2D6*
	Paroxetine	*CYP2D6*
	Pimozide	*CYP2D6*
	Sertraline	*CYP2C19*
	Trimipramine	*CYP2C19/CYP2D6*
Transplantation	Tacrolimus	*CYP3A5*

## Data Availability

PGx panel validation data are available on request.

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
