# Peer review of "Implementing Pharmacogenomics Testing: Single Center Experience at Arkansas Children’s Hospital"

_jpm, 2021, doi:10.3390/jpm11050394_

Round 1

Reviewer 1 Report

This manuscript described the PGx implementation approach in the Arkansas Children’s hospital that has several highly skilled medical professionals and expansion of clinical research capabilities. PGx is now getting therapeutic choices. It may correct the outcome of success or failure of drug therapy.

I endorse the publication of this manuscript after minor revision.

Please spelling out ”CDS” and “EPIC” in lines 78-79.

Please spelling out ”EHR” in line 74.

Please spelling out ”EMR” in line 83.

Table 1: The drugs associated with extant pediatric pharmacogenomic guidance were summarized in Table 1. The reader may make an easy understanding if associated genes were indicated in this table.

Please state the approvals of this study in the ethics committee in ACH.

Table 4. Drugs name and gene (protein? Because the letter is not.) name is not aligned.

The figure 4 and 5 are unclear. 

Author Response

We very much appreciate your suggestions and comments to improve the manuscript. We have made the changes as suggested by you. These changes are:

Please spelling out ”CDS” and “EPIC” in lines 78-79.

This has been corrected to clinical decision support (CDS)- Line 82

And EPIC is not an abbreviation and as per the company’s suggestion we have added EPICTM. -line 83

And for BPAs (EPIC © 2021 Epic Systems Corporation)-lines 289 and 322.

Please spelling out ”EHR” in line 74.

This has been corrected to electronic health record (EHR) -line 78

Please spelling out ”EMR” in line 83.

After suggestion from co-authors this has been changed to EHR-Line 88 and 100

Table 1: The drugs associated with extant pediatric pharmacogenomic guidance were summarized in Table 1. The reader may make an easy understanding if associated genes were indicated in this table.

Now the changes to Table 1 show the respective associated genes/s-Line 125

Please state the approvals of this study in the ethics committee in ACH.

This was mistake on my part, the IRB number has been added -lines 104-108.

Table 4. Drugs name and gene (protein? Because the letter is not.) name is not aligned.

Table 4 has been redrawn and aligned.

The figure 4 and 5 are unclear. 

Figures 4 and 5 has been enlarged to make it readable.

Reviewer 2 Report

Overall, this is an interesting well-written study. It highlights pediatric PGX implementation which remains not widely available. The biggest suggestion for improvement is the manuscript is missing a discussion/acknowledgement about the differences of PGX snp effects in pediatric patients vs adult patients. Most, if not all, were discovered and studied in adults with the assumption of similar relationships in pediatric patients but thus should be explicitly highlighted in the discussion.

Author Response

We thank you for your constructive suggestions and comments.

  1. This important suggestion was overlooked on my part. As per reviewer’s suggestions, I have added in discussion the changes in gene expression in adult vs pediatrics because of ontogeny. Lines 351—366. 
  1. As suggested I have added information pertaining to contributions of adult PGx studies, which have given guidance for pediatric clinical research and hence genotype guided recommendations. Lines 369-376.
